# TRAJECTORY FIRST: A CURRICULUM FOR DISCOVERING DIVERSE POLICIES

## ABSTRACT

Being able to solve a task in diverse ways makes agents more robust to task variations and less prone to local optima. In this context, constrained diversity optimization has emerged as a powerful reinforcement learning (RL) framework to train a diverse set of agents in parallel. However, existing constrained-diversity RL methods often under-explore in complex tasks such as robotic manipulation, leading to a lack in policy diversity. To improve diversity optimization in RL, we therefore propose a two-stage curriculum. The key idea of our method is to leverage a spline-based trajectory prior as an inductive bias to generate diverse, high-reward behaviors in the first stage, before learning step-based policies in the second. In our empirical evaluation, we provide novel insights into shortcomings of skill-based diversity optimization, and demonstrate empirically that our curriculum improves the diversity of the learned skills.

## 1 INTRODUCTION

Reinforcement Learning (RL) has driven breakthroughs in robot locomotion (Hwangbo et al., 2019), game-playing (Mnih et al., 2015; Silver et al., 2017), and foundation-model finetuning (Bai et al., 2022). While effective, most RL methods assume a unimodal action distribution and produce only a single policy. In contrast, humans and animals routinely solve the same task using multiple qualitatively different strategies. Such variability is also desirable in RL, as strategy diversity increases solution quality and robustness (Page, 2017; Hong & Page, 2004). Therefore, this work considers the discovery of a policy set that maximizes the reward in diverse ways.

A number of previous works have investigated this problem from various perspectives. Notably, the fields of Novelty Search (NS) and Quality-Diversity (QD) have proposed a multitude of algorithms which populate an archive of solutions based on their novelty and performance (Lehman & Stanley, 2011a;b; Conti et al., 2018). Further, gradient-based RL approaches define intrinsic diversity rewards that they combine with extrinsic task rewards using Lagrange multipliers (Zahavy et al., 2023), bandits (Parker-Holder et al., 2020), or linear combinations (Kumar et al., 2020; Masood & Doshi-Velez, 2019; Gangwani et al., 2019). While effective, the above approaches are not without shortcomings. QD may produce exceptional results, but often at the cost of sample efficiency and manual feature design. Gradient-based diversity or entropy bonuses in RL may still collapse to a few modes and remain under-evaluated in challenging contact-rich tasks (Rho et al., 2025; Emukpere et al., 2024), a finding which we corroborate in this work.

Inductive biases such as hierarchical policy structures (Pateria et al., 2021), graph-based relational representations (Battaglia et al., 2018), and physics-based priors (Ramesh & Ravindran, 2023) have driven significant advances in RL. We argue that diversity optimization also benefits from inductive biases. We propose a new and simple *trajectory-first* curriculum for learning diverse policies that explores at the level of smooth trajectories instead of neural network parameters (Section 5). Concretely, the curriculum *(i)* uses an evolutionary search over open-loop action sequences to uncover a diverse set of high-reward behaviors, and *(ii)* distills these behaviors into distinct, off-policy, model-free policies. While prior work proposed similar formulations that first solve exploration and then learning (Campos et al., 2020; Nair et al., 2018), we do not rely on human demonstrations and propose an evolutionary approach to maximize diversity at trajectory level instead of optimizing neural-network parameters, which can be inefficient. Based on our algorithm, we empirically highlight

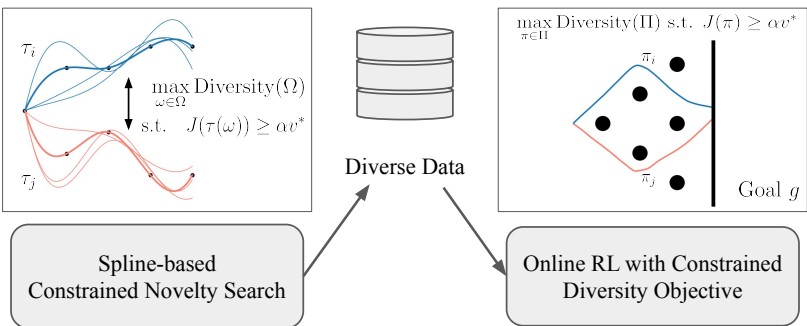

Figure 1: Overview of the proposed diversity curriculum. We use a spline-based trajectory prior to improve exploration. First, an evolution strategy explores in trajectory space to maximize novelty of trajectory parameters $\omega \in \Omega$ under performance constraints. Then, this data is used to warmstart the online training of multiple RL agents $\pi \in \Pi$ to solve the same optimization problem in policy space.

shortcomings of existing diversity optimization methods in Section 5 and illustrate how the proposed curriculum enables learning diverse sets of robot manipulation policies.

In short, we make three contributions with this work. First, we propose a novel curriculum for diversity optimization under extrinsic task rewards (see Fig. 1). Second, we introduce *Constrained Novelty Search (CNS)* to discover diverse trajectories in the first step of this curriculum (Section 3.1). Finally, we investigate how diversity can be maintained during online optimization of control policies from this data (Section 3.2).

## 2 PRELIMINARIES

**Markov Decision Process:** We model each task as a discrete-time Markov Decision Process $M = (\mathcal{S}, \mathcal{A}, p, r, \gamma)$ (Puterman, 2014, MDP). At each time step $t$, the agent in state $s_t \in \mathcal{S}$ selects action $a_t \in \mathcal{A}$, transitions to $s_{t+1}$ with probability $p(s_{t+1} \mid s_t, a_t)$, and receives reward $r_t \triangleq r(s_t, a_t) \in [r_{\min}, r_{\max}]$. The objective is to learn a policy $\pi_\theta : \mathcal{S} \times \mathcal{A} \to \mathbb{R}^+$, parameterized by $\theta \in \mathbb{R}^d$, that maximizes the discounted return $J(\pi) = \sum_{t=\ell}^{\infty} \mathbb{E}_{(s_t, a_t) \sim \pi} \left[ \gamma^{t-\ell} r(s_t, a_t) \right]$ with discount factor $\gamma \in [0, 1)$. We denote by $\rho_\pi(s, a)$ the state–action occupancy measure and by $\rho_\pi(s)$ its marginal over states following Haarnoja et al. (2018).

**Constrained Diversity Optimization:** While earlier works used scalars to balance diversity and task rewards, Zahavy et al. (2023) introduced the following constrained MDP formulation:

$$\max_{\Pi^n} \text{Diversity}(\Pi^n) \qquad \text{s.t. } J(\pi) \geq \alpha v^*, \quad \forall \pi \in \Pi^n \tag{1}$$

where $\Pi^n$ is the current set of policies, $v^* \triangleq \max_{\pi \in \Pi} J(\pi)$ is the value of the optimal policy and $\alpha \in [0, 1)$ is a hyperparameter controlling the optimality constraint. This constrained optimization problem can then be solved efficiently using Lagrange multipliers that are tuned using dual ascent (Altman, 2021; Borkar, 2005). We adopt the same problem formulation in this work. To quantify diversity, we will measure the distance to the nearest neighbor, which shall be maximized:

$$\text{Diversity}(\Pi^n) := \frac{1}{n} \sum_{i=1}^{n} \min_{\pi_j \neq \pi_i} \| \mathbb{E}_{s \sim \rho_{\pi_i}} [\phi(s)] - \mathbb{E}_{s \sim \rho_{\pi_j}} [\phi(s)] \|^2, \tag{2}$$

where $\phi(\cdot) : \mathcal{S} \to \mathbb{R}^f$ are state-based features, which generally can be manually defined or learned, for instance using the successor feature method (Barreto et al., 2017; Abbeel & Ng, 2004). To optimize Eq. (1), the common framework is to employ one-hot skill encodings $z(s_t) \in [0, 1]^n$ as conditioning for a single policy and $Q$-function when learning (Eysenbach et al., 2019; Zahavy et al., 2023). We follow this approach in our work and slightly abuse notation in using $z(s_t)$ to indicate the skill of a state and $z(\tau)$ for the skill of a full trajectory.

**Novelty Search:** Most novelty-based approaches to skill learning maximize the entropy of the policy set by using the entropy of the current policy set as an intrinsic reward (Conti et al., 2018; Lehman & Stanley, 2011a; Liu & Abbeel, 2021). To quantify entropy, the particle-based entropy estimator by Singh et al. (2003) is commonly employed which estimates the sparsity of the distribution based on the distance between the datapoints $\{x_i\}_{i=1}^n$ and their $k$-nearest neighbor: $\mathcal{H}_{\text{particle}}(X) \propto \sum_{x_i \in X} \log \|x_i - x_i^{(k)}\|$. In practice, we choose $k = 1$. In other words, we measure the particle-based entropy as distance to the nearest neighbor, and add a constant $c = 1$ for numerical stability:

$$\mathcal{H}_{\text{particle}}(X) := \sum_{x_i \in X} \log \left( c + \min_{x_j \neq x_i} \|x_i - x_j\| \right). \tag{3}$$

## 3 EVOLUTIONARY EXPLORATION FOR DIVERSE POLICY DISCOVERY

Solving the problem in Eq. (1) requires an initialization that is sufficiently diverse to prevent the diversity optimization from only occurring locally. Since the policies are initialized randomly at the beginning, we find that the discovered policy set rarely covers the task space sufficiently and instead focuses on a subset of greedy solutions. While shaping the extrinsic reward is an option to encourage diverse interactions between agent and environment, tuning such reward functions is a tedious task. We propose an alternative approach to this, which $(i)$ uses an evolution strategy (ES) to explore in the space of trajectories before $(ii)$ learning a diverse set of policies from this data. We provide an intuition for this approach in Fig. 1 and describe both stages of our curriculum in the following. For a formal algorithmic description of the method, we refer to Appendix A.

### 3.1 CONSTRAINED NOVELTY SEARCH FOR SPLINE-BASED EXPLORATION

The first stage of our curriculum directly optimizes agent trajectories $\tau \in \mathbb{R}^{T \times u}$, where $T$ denotes the number of timesteps and $u$ the robot's degrees of freedom. We represent a trajectory as a B-spline parameterized by a control point matrix $\omega \in \mathbb{R}^{m \times u}$. Following the constrained diversity optimization objective in Eq. (1), we optimize a set of trajectory parameters $\Omega = \{\omega_i\}_{i=1}^n$ such that the resulting trajectories $\tau(\omega)$ are as diverse as possible under the constraint of near-optimality:

$$\max_{\Omega^n} \text{Diversity}(\Omega^n) \qquad \text{s.t. } J(\tau(\omega)) \geq \alpha v^*, \quad \forall \omega \in \Omega^n, \tag{4}$$

where $J(\tau(\omega_i)) = \sum_t r_{\text{ext}}(s_t^i, a_t^i)$, and $v^* = \max_\omega J(\tau(\omega))$. Since the extrinsic task reward function $r_{\text{ext}}$ is generally non-differentiable we optimize this problem using a novel multi-population evolution strategy (ES) which we describe in the following. Following Zahavy et al. (2023) we solve the dual optimization using gradient ascent with bounded Lagrange multipliers $\{\lambda_i\}_{i=1}^n$, which yields the following reward that we evaluate on each trajectory $\tau_i = \tau(\omega_i)$:

$$r(\tau_i) = (1 - \sigma(\lambda_i)) \, r_{\text{int}}(\tau_i) + \sigma(\lambda_i) \, r_{\text{ext}}(\tau_i) \tag{5}$$

$$= \sum_{s_t \in \tau_i} (1 - \sigma(\lambda_i)) \, \mathcal{H}_{\text{particle}}(\phi(s_t)) + \sigma(\lambda_i) \, r_{\text{ext}}(s_t) \tag{6}$$

$$= \sum_{s_t^i \in \tau_i} \left[ [1 - \sigma(\lambda_i)] \log \left( 1 + \min_{\tau(\omega_j) \in \Omega} \|\phi(s_t^i) - \phi(s_t^j)\|_2 \right) + \sigma(\lambda_i) \, r_{\text{ext}}(s_t) \right], \tag{7}$$

where $\phi$ is a feature extraction function that projects the states to a lower dimension, $\lambda_i$ is the $i$-th Lagrange multiplier, and $\sigma$ denotes the sigmoid function $\sigma(x) = 1/(1 + \exp(-x))$. We note that Eq. (7) is a generalized version of the novelty search objective from Conti et al. (2018), but using population-level Lagrange multipliers instead of a single heuristically selected scalar. Using Lagrange multipliers permits not only to consider different weights for different populations, but also a dynamic adaptation of these weightings. We denote this objective and its optimization as *Constrained Novelty Search (CNS)* in the following. While Zahavy et al. (2023) derive the analytical gradient of Eq. (2) for their intrinsic reward, CNS approximates this gradient by stochastic sampling from an ES, which approximates natural gradient steps on the novelty objective (Akimoto et al., 2012; Glasmachers et al., 2010; Hansen & Ostermeier, 2001).

Similar to prior work (Zahavy et al., 2023; Faldor et al., 2025), we introduce state features $\phi : \mathcal{S} \to \mathbb{R}^f$ to avoid relying on the manually defined descriptors that Conti et al. (2018) use. While most of

this prior work learn such feature mappings, we use a fixed random projection $\phi(x) = Qx$ where $q_i \sim \mathcal{N}(0, I)$ are the basis vectors of the projection $Q \in \mathbb{R}^{f \times S}$ that we sample from a standard normal distribution. Using these representations comes with two benefits. First, the blackbox optimization is stabilized since the intrinsic objective is defined on a stationary embedding. Second, the approximation error of the feature distances is bounded following the Johnson-Lindenstrauss lemma (Johnson et al., 1984). To improve the stability of the optimization we further use an exponentially-moving-average (EMA) to low-pass filter the expected features per trajectory.

Finally, we note again that we optimize in $\mathbb{R}^{m \times u}$ instead of $\mathbb{R}^d$ at this step where $d$ is the dimension of policy parameters. Since typically $m \times u \ll d$, we can optimize Eq. (7) with the CMA-ES (Hansen & Ostermeier, 2001), which is empirically more sample efficient in moderately high dimensional parameter spaces than using isotropic search distributions (Conti et al., 2018; Parker-Holder et al., 2020). We will revisit this difference in the experiment section where we compare our method to parameter exploration.

## 3.2 Efficient Policy Diversity Optimization from Prior Data

Given a diverse dataset $\mathcal{D} = \{(\tau_i, z_i, r_{\text{ext}}(\tau_i))\}_{i=1}^{\ell}$ with skill labels $z_i \in \{1, \ldots, n\}$ from CNS, our second stage learns $n$ reactive, skill-conditioned policies that preserve diversity while satisfying the near-optimality constraint. We build on the off-policy Domino framework (Zahavy et al., 2023), and augment it with three key modifications inspired by efficient offline-to-online RL (Ball et al., 2023).

Following Zahavy et al. (2023), we use the gradient of the diversity objective as intrinsic reward for policy optimization, that is

$$r_{\text{int}}(s_t, a_t \mid z) = \phi(s_t)^\top \left( \bar{\phi}_z - \bar{\phi}_j \right), \qquad \text{s.t.} \quad \bar{\phi}_j = \arg\min_{j \neq z} \|\bar{\phi}_z - \bar{\phi}_j\|_2^2, \tag{8}$$

where $\bar{\phi}_z = \mathbb{E}_{s \sim \rho_{\pi_z}}[\phi(s)]$ are the expected features per skill. For consistency, the intrinsic term measures novelty in the same projection space $\phi$ used in CNS. We balance extrinsic and intrinsic rewards as in Eq. (7) using bounded Lagrange multipliers that we update via dual ascent to enforce the near-optimality constraint. Similar to Domino, we track a running estimate of the expected features per skill, which we initialize with the population means from CNS. Since Domino leaves the first skill unconstrained, we estimate the per-skill performances from the undiscounted trajectory returns in the CNS buffer and choose the best skill as the leader skill.

To efficiently incorporate the CNS data, we use symmetric sampling (Ball et al., 2023; Vecerik et al., 2017; Ross & Bagnell, 2012), which means that each batch is composed of equal parts of online and offline transitions. Unlike prior work, we do not use the full data, however, since we found that trajectories from early iterations of CNS might fail to make meaningful interactions with the environment. To also exploit suboptimal data, we use the top 50% trajectories for each skill that were collected after an initial ES burn-in period. When learning online with this data, we stratify each offline batch per skill to guarantee that no skill is undersampled. We find that this stabilizes the optimization, especially in domains in which the behavioral modes tends to be close.

Further, we employ a high number of learning steps per environment step (update-to-data ratio, UTD) to learn from the diverse CNS data efficiently. This approach accelerates the propagation of exploration signals through the critic network but requires strong regularization to avoid overfitting. To address this, we follow Ball et al. (2023) in using random ensemble distillation (Chen et al., 2021), and observation and layer normalization. Prior work typically achieves a high UTD by updating the critic more frequently than the policy, but we deviate from this practice because our method updates Lagrange multipliers alongside the policy and critic, introducing a second source of variance into the policy target. Although updating the policy less often than the critic can help stabilizing intrinsic and extrinsic values, we found that synchronizing Lagrange multiplier and policy updates destabilizes training. Even with EMAs of per-policy values, the constrained optimization can become unstable due to a high variance in policy targets. Therefore, we opt to update the policy several times per environment step, effectively holding the multipliers fixed across these updates and thus greatly reducing target variance. We still update the critic multiple times per policy step to speed up learning, but we find that distributing the UTD more evenly between policy and critic updates yields the most efficient and stable learning. As a rule of thumb, we recommend nesting $\sqrt{\text{UTD}}$ policy updates in just as many critic updates.

# 4 RELATED WORK

**Diversity-Driven Policy Discovery.** Various methods to search diverse policies have been proposed. Quality-Diversity (QD) and evolutionary methods search in a gradient-free manner, populating archives of high-performing, behaviorally distinct solutions (Mouret & Clune, 2015; Cully et al., 2015) or co-optimizing fitness and novelty across populations (Ulrich & Thiele, 2011; Parker-Holder et al., 2020; Conti et al., 2018; Vassiliades et al., 2017; Braun et al., 2025). In principle, these approaches could be used in the first step of the proposed curriculum. However, most prior work focuses on optimizing in policy parameter space before distilling policies (Faldor et al., 2023; Macé et al., 2023; Chalumeau et al., 2023), a less effective process as we find in this work. Recently, gradient-based RL has been reformulated to discover multiple policies via intrinsic diversity bonuses. Notably, Eysenbach et al. (2019) maximize mutual information between skills and states, but focus on unsupervised skill discovery, while we target task-driven diversity in this work. For environments with extrinsic rewards, prior methods either learn policies sequentially (Fu et al., 2023; Masood & Doshi-Velez, 2019; Zhou et al., 2022; Chen et al., 2024), or in parallel for greater efficiency (Zahavy et al., 2023; Gangwani et al., 2019; Chen et al., 2024; Celik et al., 2024). While earlier works balance extrinsic and intrinsic rewards with fixed scalars (Masood & Doshi-Velez, 2019; Liu et al., 2017), more recent works proposed adaptive weighting schemes (Parker-Holder et al., 2020; Kumar et al., 2020). In particular, Zahavy et al. (2023) proposed constrained optimization with Lagrange multipliers, which we adopt in this paper.

**Exploration in RL.** Exploration is a fundamental aspect of RL, enabling agents to effectively sample the environment, avoid premature convergence to suboptimal policies, and enhance both learning performance and generalization. Accordingly, numerous exploration strategies have been proposed in the literature (Ladosz et al., 2022). A common strategy perturbs the agent's actions – often via Gaussian or temporally correlated noise processes (Fujimoto et al., 2018; Hollenstein et al., 2022). Another line of work introduces parameter noise, where noise is applied directly to the agent's parameters rather than to its actions (Plappert et al., 2018; Fortunato et al., 2018). Beyond pure noise, intrinsic-reward methods augment the extrinsic task reward with bonuses for novelty. These approaches include techniques based on knowledge-based exploration, which maximizes prediction error (Burda et al., 2019), competence-based exploration (Houthooft et al., 2016; Eysenbach et al., 2019; Laskin et al., 2022; Zheng et al., 2024) and data-based exploration (Liu & Abbeel, 2021). While the above intrinsic exploration objectives or parameter noise explore in the policy space, we propose to perform intrinsically motivated exploration in the much more structured trajectory space. By optimizing a novelty objective on trajectories using an ES, we can explore over entire behaviors, much like temporally correlated action noise, but with a self-optimizing noise distribution. While we are not the first to investigate RL at trajectory level (Otto et al., 2023; Klink et al., 2020; Celik et al., 2024), these works do not learn step-based reactive policies but predict full action sequences, which is a drawback in practical domains such as robotics. Finally, unlike methods that rely on expert demonstrations to guide exploration (Nair et al., 2018; Salimans & Chen, 2018), our approach does not require prior knowledge or external supervision.

**RL Finetuning from Datasets.** Offline RL addresses the issue of data inefficiency inherent in online RL by training solely on a fixed dataset of past interactions, but it often suffers from sub-optimality due to limited or biased data (Liu et al., 2024). To overcome these problems, the paradigm of offline-to-online RL has been proposed, where a policy is trained on offline and online data in conjunction.The approaches broadly fall into two categories based on whether the dataset from the replay buffer is discarded after a pretraining phase, or whether it is retained (Zhou et al., 2025). Pretrain-and-discard methods first pretrain a policy or critic on the dataset and then use the same networks for finetuning without retaining any prior data (Zhou et al., 2022; Uchendu et al., 2023; Wolczyk et al., 2024). Data retention methods keep the dataset in the replay buffer for at least a fraction of the training procedure. Notably, recipes that train a policy online, mixing samples from the prior dataset and current rollouts have been shown to be surprisingly effective while being easy to implement (Ball et al., 2023; Vecerik et al., 2017; Nair et al., 2018). We adopt this mixing strategy for our CNS dataset finetuning but observe that none of these approaches explicitly address the preservation or enhancement of policy diversity during online adaptation. In this work, we close that gap by investigating how offline–online data selection and mixing influence the retention and amplification of diverse behaviors.

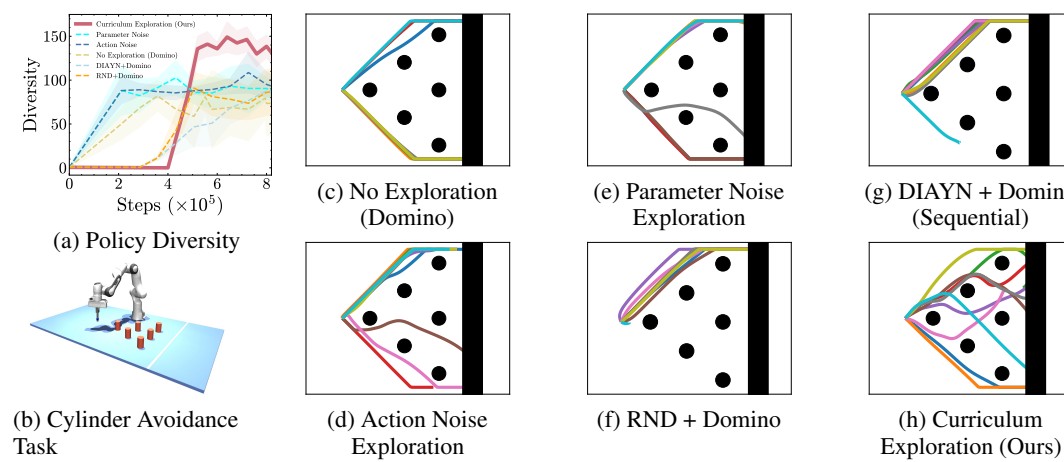

(a) Policy Diversity

(b) Cylinder Avoidance Task

(c) No Exploration (Domino)

(d) Action Noise Exploration

(e) Parameter Noise Exploration

(f) RND + Domino

(g) DIAYN + Domino (Sequential)

(h) Curriculum Exploration (Ours)

Figure 2: Qualitative results for the cylinder avoidance task. Each plot depicts $xy$ trajectories around black obstacles. Each of the 10 skills is plotted in a separate color. Curriculum exploration is the only method that finds all collision free paths through the maze.

## 5 EXPERIMENTS

Our experiments are designed to provide insights into the shortcomings of naively applying diversity constraints and how these can be mitigated. As such, we aim to answer the following questions: **(a)** Does constrained diversity optimization sufficiently explore the environment? **(b)** Does a curriculum with constrained novelty search increase policy diversity in skill learning? **(c)** Do we need diversity objectives and performance constraints?

**Environments.** We conduct our experiments focusing on environments from robotics. We use three environments for evaluation: (a) a cylinder avoidance task where a rod attached to a robot gripper must be navigated through a maze without collisions (see Fig. 2). The agent is rewarded for minimizing the distance to the goal line and penalized for touching obstacles. This task enables a rich set of behaviors since many paths through the maze are possible. To avoid the trivial solution of moving over the top of all obstacles, we fix the $z$-position of the rod and use $xy$-endeffector position control. We train 10 different skills for this task. (b) Second, we use a cube pushing task in which an 8-dof Franka Panda robot is tasked to push a cube as far away from the center of a table as possible. This task enables a rich set of behaviors since there are no contact encouraging terms in the reward, so any object manipulation is conceivable. We train 5 skills for this environment and measure diversity in terms of cube positions. (c) Third, we use a button pressing task in which a Franka Panda robot is tasked to press a button that is placed on a table in front of him. We train 5 skills for this environment and measure diversity in terms of contact forces between the robot links and button frame. For full environment details, we refer to Appendix B.

**Baselines.** We compare our method to five baselines: (a) We use plain diversity optimization with optimality constraints, i.e. **Domino**, without additional exploration, to investigate how well such methods explore. Second, we compare against two methods for noise-based online exploration: (b) Gaussian **action noise** $\epsilon \sim \mathcal{N}(0, I)$ as a non-diversity-specific exploration method as well as (c) **parameter noise** exploration (Fortunato et al., 2018). In particular, the latter is an interesting baseline to our method since evolutionary optimization is commonly employed to explore in parameter space. (d) We compare against two alternative methods to generate a diverse set of trajectories for downstream training: we compare against **DIAYN** (Eysenbach et al., 2019) as a representative for skill-based exploration methods in RL. Following the common procedure, we first use DIAYN for unsupervised exploration and then finetune the resulting policy using Domino to investigate whether unsupervised skill-based RL methods can be used to discover efficient robot manipulation skills. (e) The second sequential method that we compare against first uses random network distillation (**RND**) (Burda et al., 2019) to generate diverse trajectories, which we then cluster and use as an offline buffer for downstream learning. This approach is inspired by Li et al. (2024) who iteratively use RND to discover diverse behaviors.

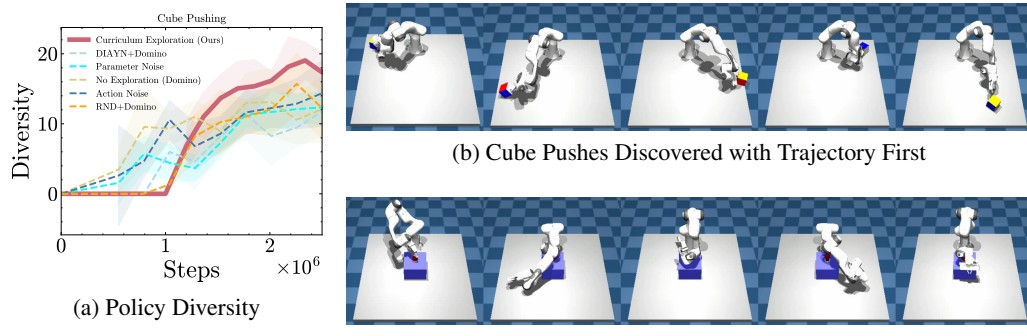

(a) Policy Diversity

(b) Cube Pushes Discovered with Trajectory First

(c) Button Press Strategies Discovered with Trajectory First

Figure 3: Qualitative results for the cube pushing task. Curriculum exploration produces creative button press strategies that explore the contact force space. Quantitative results depict 95% CI and IQM across 6 seeds.

**Setting.** We implement the baseline diversity optimization as described in Domino (Zahavy et al., 2023), and base our code on the public implementation thereof (Grillotti et al., 2024). As stated above, all code is based on SAC (Haarnoja et al., 2018), and uses observation normalization, critic ensembling, and layer norm. For further implementation details, we refer to Appendix B. For evaluation we report the mean return across all policies across 6 seeds as well as the mean diversity defined by Eq. (2), but using ground truth state information of the rollout instead of the featurized state representations for the diversity metric computation in our evaluation. Following prior work, we use the minimum spanning tree (MST) diversity score to measure diversity and exploration quality with an edge weight of 1 (Toussaint et al., 2024; Papenmeier et al., 2025). That is, we construct a construct an MST from the generated trajectories and measure its size as the sum of all edges in the tree. This metric captures how spread out the trajectories are.

**Q1. Does constrained diversity optimization sufficiently explore the environment?**

To answer this question, we look at the performance of running *diversity optimization without any additional exploration mechanism*. We observe in Fig. 2 that the diversity optimization algorithm Domino underexplores the domain, which leads to little behavioral diversity. While the method successfully solves the problem, only a few behavioral modes are found. This highlights that learning a set of diverse skills requires the discovery of a sufficient number of behaviors in the first place. Further, our quantitative results (Fig. 4) highlight that established single-skill exploration methods only partly resolve this issue, as the overall diversity of the solutions remains limited, which is also visible in Fig. 2. We observe that action noise exploration yields good performance in terms of returns on the investigated tasks, but it only slightly improves diversity on the cube pushing task. Further, we observe that unsupervised exploration methods like DIAYN are insufficient to learn multiple performant and diverse skills. While they have been known to excel at discovering multiple strategies from *which at least one* tends to be useful on downstream tasks, this offers only limited support if the downstream optimization requires training multiple diverse policies. Although such methods can be combined with extrinsic rewards, prior work has demonstrated that using Lagrange multipliers to automatically balance diversity and task reward are superior to scalar combinations (Zahavy et al., 2023).

**Q2. Does a curriculum with constrained novelty search increase policy diversity?**

To answer this question, we consider the results of training agents using our proposed curriculum (Section 3). As we can see in Fig. 2, using the curriculum enables the discovery of almost all paths though the maze. Other methods like Domino and DIAYN tend to explore more subtle variations such as the length of the path, while other methods such as RND suffer from skills collapsing to single modes. These qualitative findings are corroborated quantitatively in Fig. 4, which shows that the proposed curriculum produces the most diverse policies on the maze and button tasks. For the cube task, the quantitative difference in diversities is less pronounced than for the other tasks although our method performs still well on this task (Fig. 3). We observe that the baselines produce longer trajectories than our method which result in greater distances, even if the push directions are

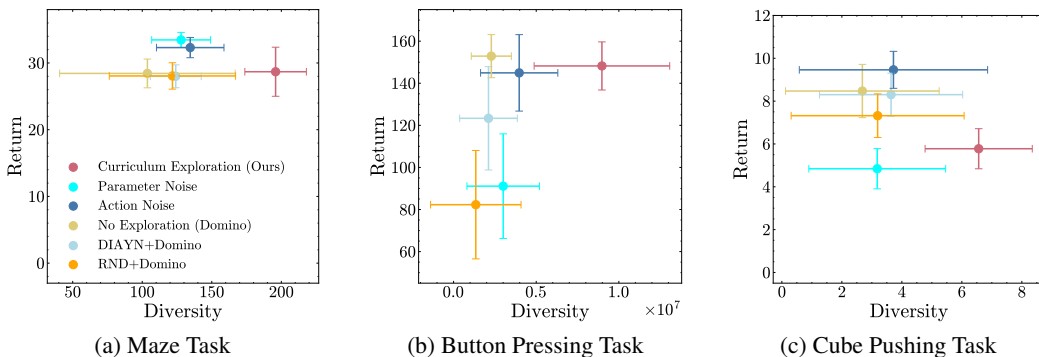

(a) Maze Task        (b) Button Pressing Task        (c) Cube Pushing Task

Figure 4: Quantitative evaluation. Curriculum exploration leads to higher policy diversity at high performance on the maze and button pressing tasks. We report interquartile mean (IQM) and 95% confidence intervals across 6 seeds.

similar. Yet, our method still produces the most diverse solutions as depicted in Fig. 3a. While the policy performances of our method are lower on the latter task, they are still within the $\alpha = 0.5$ optimality bound that we used in our experiments for this task.

### Q3. What is the value of diversity objectives?

This question aims to investigate whether the diversity objective (Eq. (1)) during RL is needed given diverse data from the first stage of the curriculum. To answer this question, we run constraint-free population-based training (Jaderberg et al., 2017, PBT) based on the CNS data, but only maximizing policy return. Our results in Fig. 5b show that using the diversity objective indeed increases the policy diversity. At the same time, we observe a slight decrease in task performance when optimizing for diversity, which we explain by the constraint from Eq. (1), which permits a certain amount of slack $\alpha$. We believe that these observations provide valuable insights into RL finetuning for diversity. As stated above, prior finetuning literature neglected diversity, and we believe that the presented recipe closes this gap. Further, we investigate whether the inclusion of the Lagrange multiplier in the CNS formulation improves the evolutionary trajectory optimization in comparison to the scalarized novelty search formulation from previous work (Conti et al., 2018). Fig. 5a displays the kNN population entropy across novelty search iterations. We observe that the sample efficiency for high entropy is much higher for the proposed CNS objective, reaching higher entropy states in much less iterations compared to the scalarization. This demonstrates that the generalized formulation of novelty search may be a valuable tool for exploration and discovery in the future, also beyond RL.

### Q4. How to train diversity constrained policies?

Last, we investigate the training dynamics of our method. As noted by Ball et al. (2023), the update-to-data ratio (UTD) plays a crucial role when learning from prior data. While most prior work (Chen et al., 2021; Ball et al., 2023) achieves fast convergence by updating the critic dramatically more often than the policy, we find that the training dynamics of our approach slightly differ since the Lagrange multipliers introduce another source of target nonstationarity into the policy training. To better understand offline-to-online RL in the constrained diversity setting, we therefore compare three policy update strategies for a fixed UTD=16 in Fig. 5c. *Low delay* refers to updating the policy at every critic update step, while *high delay* refers to updating the policy only once per environment step, i.e. the RLPD setting from Ball et al. (2023). *Medium delay* represents our recommendation: splitting the UTD more evenly by nesting multiple critic updates into multiple actor updates. Our results show that using a high delay, however, destabilizes training which is reflected in the oscillating training curves and lower data efficiency. Conversely, using a low delay leads to the worst sample efficiency. We find that splitting the UTD into nested policy and critic updates stabilizes training and leads to the highest sample efficiency. Using this strategy, the Lagrange multipliers can effectively be fixed for multiple policy steps, reducing variance in the targets without overfitting to a varying critic.

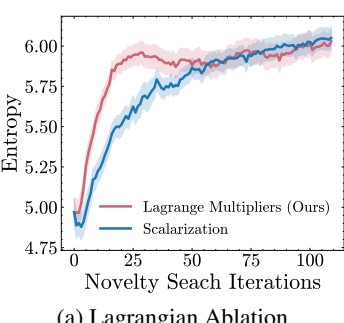 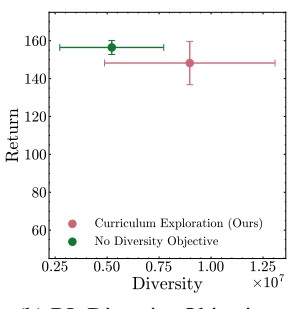 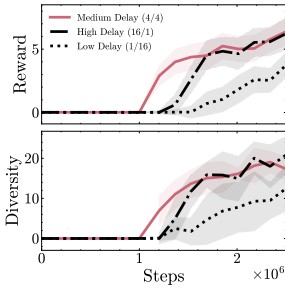

(a) Lagrangian Ablation  (b) RL Diversity Objective  (c) Policy Delay Ablation
Ablation

Figure 5: Ablation studies. (a) We ablate the usage of Lagrange multipliers and the update-to-data ratio (UTD) when training the policies. (b) We compare performances for RL with diverse data from CNS, but no diversity objective during training (PBT) against our proposed approach. (c) We analyze the role of the update-to-data ratio (UTD) during policy optimization, reporting different ratios of (Critic steps / Actor steps) per environment step. We report interquartile mean (IQM) and 95% confidence intervals across 6 seeds.

## 6 CONCLUSION & DISCUSSION

We introduced a two-stage, trajectory-first curriculum that can discovering diverse skills in challenging robotic domains. In the first phase, we use a constrained novelty search evolution strategy to explore trajectories. In the second stage, we train a set of diverse reactive control policies given the CNS data. We have shown in our experiments that: **(a)** naïvely applying constrained diversity objectives in policy space leads to under-exploration and thus fails to discover truly diverse skills (Fig. 2). **(b)** By first exploring diverse trajectories using constrained novelty search, the diversity optimization can be improved (Fig. 4). **(c)** Using proper performance constraints during evolutionary novelty search improves efficiency over scalar formulations (Fig. 5b). **(d)** Using diverse data for policy training alone does not guarantee truly diverse policies, but diversity objectives are still required to maintain full diversity (Fig. 5a). **(e)** Updating the policy multiple times per Lagrange multiplier step, but less often than the critic is crucial for efficient offline-to-online constrained diversity optimization.

Despite these advances, our approach is not without limitations. CNS exploration does not exchange extrinsic reward information between populations. Therefore it may not exploit representation sharing mechanisms as neural networks do, leading to potentially lower sample efficiency when a high number of skills, e.g. 100, shall be learned. Nevertheless we find that this is also a strength in disguise since this also prevents different skills leaking information into each other. Further, we note that our current study only investigates deterministic environments to optimize the trajectories. Future work could address this by using CNS over multiple initializations that are warmstarted from the previously found solution. Last, while we observe that on most tasks our method does not mitigate task performance, this is the case on the cube pushing task. We find that this is partly due to the sensitivity of the approach to the optimality threshold $\alpha$. While lower values tend to work better for CNS, higher values might be required to guarantee better policies. Future work could therefore explore using different thresholds for the different stages of the curriculum.

Our method enables to explore in the full observation space, but also on subsets thereof, i.e., on selected features. This provides a promising avenue for research, in particular in robotics to discover novel robot manipulations that explore contact forces or other parts of the state space. We aim to extend our research by investigating which features provide good learning signals for robot manipulation discovery.

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

# A  ALGORITHM

---

**Algorithm 1** Curriculum for discovering diverse policies.

---

**Input:** Environment $env$, Optimality ratio $\alpha$, Num. skills $n$, Learning rates $\kappa_\lambda, \kappa_\xi, \kappa_V, \kappa_\pi$, Init. SAC temperature $\xi$

1: **// 1. Constrained Novelty Search**
2: Initialize population parameters $\omega_i$ for skills $i = 1, \ldots, n$
3: $\mathcal{D}_{cns} \leftarrow \{\}$
4: **for** iteration $t = 1, \ldots, T$ **do**
5:    **for** Population $i = 1 \ldots n$ **do**
6:       $\{\tau_1^i, \ldots \tau_m^i\} \leftarrow$ env.rollout$(\omega_i)$
7:       Update $\omega_i$ given $(1 - \sigma(\lambda_i^{cns}))\, r_{\text{int}}(\{\tau_1^i, \ldots \tau_m^i\}) + \sigma(\lambda^{cns})\, r_{\text{ext}}(\{\tau_1^i, \ldots \tau_m^i\})$    (Eq. 4)
8:       $v_i \leftarrow v_i + \kappa_\lambda\, \mathbb{E}[r_{ext}(\{\tau_1^i, \ldots \tau_m^i\})]$            (Estimate population values)
9:       **if** t % LAMBDADELAY **then**
10:          $\lambda_i^{cns} \leftarrow \lambda_i^{cns} - \alpha_\lambda \nabla(\lambda^{cns}(v_i - v^*\alpha))$
11:       **end if**
12:       $\mathcal{D}_{cns} \leftarrow \mathcal{D}_{cns} \cup \{\tau_1^i, \ldots \tau_m^i\}$
13:    **end for**
14: **end for**
15:
16: **// 2. Constrained RL Diversity Optimization**
17: **for** iteration $t = 1, \ldots, I$ **do**
18:    **// Environment steps**
19:    $z \sim p(z)$
20:    $a_t \sim \pi(a_t \mid s_t, z)$
21:    $s_{t+1} \sim p(s_{t+1} \mid s_t, a_t, z)$              (Step environment)
22:    $\mathcal{D}_{rl} \leftarrow \mathcal{D}_{rl} \cup \{(s_t, a_t, r(s_t, a_t), \phi(s_t, a_t), s_{t+1}, z)\}$
23:    **// Training steps**
24:    $\mathcal{D}_{batch} \leftarrow \mathcal{D}_1 \cup \mathcal{D}_2$ with $\mathcal{D}_1 \sim \mathcal{D}_{cns}, \mathcal{D}_2 \sim \mathcal{D}_{rl}$        (Symmetric sampling)
25:    $\xi \leftarrow \xi - \kappa_\xi \nabla J_\xi(\xi)$           (Update SAC temperature)
26:    $\lambda^{rl} \leftarrow \lambda^{rl} - \kappa_\lambda \nabla J_\lambda(\lambda^{rl})$        (Dual ascent on Eq. 1)
27:    $\theta_V \leftarrow \theta_V - \kappa \nabla J_V(\theta_V)$             (Update critic)
28:    $\theta_\pi \leftarrow \theta_\pi + \kappa_\pi \nabla J_\pi(\theta_\pi)$             (Update policy)
29: **end for**

---

# B  EXPERIMENTAL DETAILS

Each experiment is repeated across 5 different seeds. Where applicable, we report the interquartile mean (IQM) across all 5 runs and bootstrapped 95% confidence intervals in our plots. In the following we provide details about the environments and implementation that we used in this work. All the experiments are performed on an internal cluster with eight NVIDIA A40 GPUs. We evaluate our method on two robotics tasks, which are displayed in Fig. 6. Both environments requires reasoning over objects in the scene, which is generally challenging.

## B.1  ENVIRONMENTS

**Cylinder Avoidance Task.** In this task, a robot must successfully navigate a rod that is attached to its gripper around 6 cylinder obstacles without knocking them over. The agent is rewarded for minimizing the distance to the goal line and penalized for touching obstacles. To avoid the trivial solution of moving over the top of all obstacles, we fix the $z$-position of the rod and use $xy$-endeffector position control. Therefore, the action space is $a \in [-1, 1]^2$, while the observation space are position and velocity information, i.e., $s \in \mathbb{R}^4$. We train 10 different skills for this task. The reward function that we use is the following:

$$r(s, a) = s_x - target_x - \beta \cdot \mathbb{1}_{collision}(s) + x_{max},$$

where $s_x$ denotes the $x$ position of the rod, $target_x$ is the coordinate of the goal line, while $\mathbb{1}_{collision}(s)$ is a collision checking function. Further $x_{max}$ is the maximum $x$ coordinate that

is admissible, which we use as offset to guarantee position rewards. We additionally clip the rewards to be in $[-2, 2]$ to improve training stability. We choose $\beta = 100$ for our experiments. To further simplify the task, we bound the $xy$-positions to $[-4.5, 4.5]^2$, which we implement by clipping.

**Cube Pushing Task**  In this task, a 8dof Panda robot is tasked to push a cube as far away from the center of a table as possible. The agent is only rewarded for maximizing the distance between the cube's center of mass and that of the table. This task is challenging because of the deceptive rewards in the contact-rich manipulation setting. The action space is $a \in [-1, 1]^8$, while the observation space are position and velocity information of robot and cube. We train 5 skills for this environment. The reward function is defined as follows:

$$r(s, a) = \alpha\|xyz_{cube} - xyz_{EE}\| + \beta(displacement_{current} - displacement_{previous}).$$

Here $displacement = \|xy_{cube} - xy_{table}\|$ denotes the position difference between cube and table's center of mass and $xyz_{EE}$ is the Cartesian endeffector position. We approximate $\dot{q}$ by first order finite differences as $\dot{q}(x) \approx x_{t+1} - x_t$, which we find to produce better training results than using the velocities from the Mujoco simulator that we use (Todorov et al., 2012). We choose $\alpha = 0.001$, $\beta = 10.0$.

**Button Push Task**  In this environment, a 8dof Panda robot is tasked to push a button in a buttonbox in front of it. The agent is rewarded for approaching the button with any link and for pressing the button down. The action space is $a \in [-1, 1]^8$, while the observation space are position and velocity information of robot and button as well as the contact forces between robot and button. We train 5 skills for this environment and measure diversity in the contact force spae, i.e., we measure the contact forces between each link of the robot and the button to estimate behavioral diversity.

$$r(s, a) = \alpha\|xyz_{cube} - xyz_{EE}\| + \beta(displacement_{current} - displacement_{previous}).$$

### B.2 IMPLEMENTATION

We implement all algorithms in JAX (Bradbury et al., 2018). We implement Constrained Novelty Search based on the CMA-ES implementation from evosax (Lange, 2023). The code for all RL agents is based on Domino (Zahavy et al., 2023), the public implementation thereof Grillotti et al. (2024), and the STOIX ecosystem (Toledo, 2024). We follow Zahavy et al. (2023) in the choices of all hyperparameters for Domino with exceptions detailed below. For all baselines, we use ground truth observations as state features, since they are low-dimensional and should thus yield the best performances (Zahavy et al., 2023; G Leon et al., 2024). For CNS, we use the aforementioned random projections since they are an elementary part of the method. For a full list of hyperparameters, we refer to Table 1.

**Initialization**  Since we initialize Domino from prior data, we adapt the initialization. The running estimates of the state features are not initialized with $\phi^{avg} = \bar{1}/f$ for features in $\mathbb{R}^f$. Instead, we use the mean of the maximum likelihood solutions from the CNS. In other words, for each CMA-ES

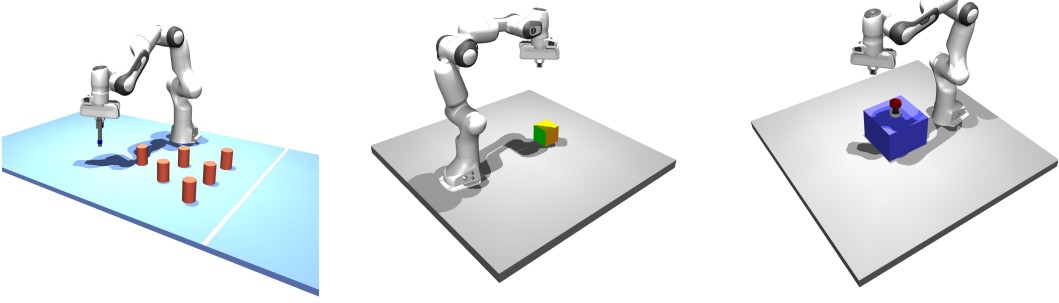

Figure 6: Considered environments. Left: cylinder avoidance task. Middle: cube pushing. Right: button pressing.

population that we run, we select the resulting trajectory parameters, roll out an additional trajectory from them and use the expected features over this trajectory as initial estimate of the features per skill. We find that this initialization is provides better results than the uniform initialization from Domino. For the values however, we follow Domino in using a zero initialization for all skills instead of using the expected reward from the final parameter rollouts. This is because such an initialization would overestimate the capacities of the current policies and thus only optimize diversity from the very beginning of training following Eq. 1.

**Network Architectures**  As stated in Section 3.1 we follow design choices from prior work in using layer normalization and observation normalization. We use the same architectures for actor and critic networks, however we perform critic ensembling for regularization and use separate heads for extrinsic and intrinsic values. All networks are optimized using Adam (Kingma & Ba, 2015).

**Constrained Novelty Search.**  We implement CNS based on the CMA-ES implementation in evosax Lange (2022). Before combining extrinsic and intrinsic fitness, we normalize both values within each subpopulation. Further, we use simple gradient descent to update the Lagrange multipliers. For higher optimization stability, we only update these parameters every iteration, but then perform 200 steps of gradient descent. To prevent gradient saturation due to the usage of sigmoids on the Lagrange multipliers, we bound them to make sure that they remain in a reasonable range. Similar to Domino, we also fix the first Lagrange multiplier to 1, so we can estimate $v^*$ based on this population. In practice, we found it more stable, however, to choose $v^* = \max_{\omega \in \Omega} v(w)$. Note that we follow this choice only during CNS.

## C  THE USE OF LARGE LANGUAGE MODELS

We used large language models (LLMs) to assist with polishing parts of the writing in this paper, e.g. to generate this statement. Additionally, we used LLMs to generate small code snippets, such as for data handling, plotting, and debugging. All conceptual ideas, methodological choices, and contributions presented in this work are entirely our own.

|  | **Cylinder Avoidance** | **Cube Pushing** | **Button Pressing** |
|---|---|---|---|
| *Environment Details* | | | |
| Observation size | 4 | 25 | 49 |
| Action size | 2 | 8 | 8 |
| Episode length | 100 | 200 | 200 |
| Num. env. steps | 800 000 | 2 500 000 | 2 000 000 |
| Num. skills | 10 | 5 | 5 |
| Optimality ratio $\alpha$ | 0.75 | 0.5 | 0.5 |
| *RL Parameters* | | | |
| Policy update freq. (data-based) | 4 | 4 | 4 |
| Critic update freq. (data-based) | 4 | 4 | 4 |
| Policy update freq. (data-free) | 1 | 1 | 1 |
| Critic update freq. (data-free) | 2 | 2 | 2 |
| Discount | 0.97 | 0.99 | 0.99 |
| Batch size | 256 | 256 | 256 |
| $\mathcal{H}_{target}$ | $\dim \mathcal{A}/2$ | $\dim \mathcal{A}/2$ | $\dim \mathcal{A}/2$ |
| Critic hidden depth | 2 | 2 | 2 |
| Critic hidden size | 512 | 512 | 512 |
| Actor hidden depth | 2 | 2 | 2 |
| Actor hidden size | 512 | 512 | 512 |
| Learn. Rate Critic | 3e-4 | 3e-4 | 3e-4 |
| Learn. Rate Actor | 3e-4 | 3e-4 | 3e-4 |
| Learn. Rate Temperature | 3e-4 | 3e-4 | 3e-4 |
| Learn. Rate Lagrange | 1e-3 | 1e-3 | 1e-3 |
| Optimizer | Adam | Adam | Adam |
| Polyak weight | 0.005 | 0.005 | 0.005 |
| Num. critics | 10 | 10 | 2 |
| Critic subset size | 2 | 2 | 2 |
| EMA weight $\alpha_\phi$ | 0.9999 | 0.9999 | 0.9999 |
| EMA weight $\alpha_v$ | 0.99 | 0.99 | 0.99 |
| *CNS Parameters* | | | |
| Num. iterations | 100 | 110 | 110 |
| Subpopulation size | 4 | 8 | 8 |
| Elite ratio | 0.5 | 0.4 | 0.4 |
| Init. $\sigma$ | 0.8 | 0.5 | 0.5 |
| Random feature dim. | 2 | 2 | 8 |
| Num. spline controls | 10 | 6 | 6 |

Table 1: Full Hyperparameter Overview

