# OpenReview forum: "Trajectory First: A Curriculum for Discovering Diverse Policies"
_ICLR.cc/2026/Conference — ICLR 2026 Conference Withdrawn Submission_

### Official Review · Reviewer_p23b · 2025-10-28

**Soundness:** 2
**Presentation:** 3
**Contribution:** 3
**Rating:** 4
**Confidence:** 3

**Summary:**

This paper proposes learning diverse skills based on a constrained diversity objective. Notably, the diversity maximization follows from optimizing the parameters of an open-loop trajectory that enables exploration on a broader state space. Subsequently, a closed-loop policy is distilled from the learned trajectories to embed the diversity into a high-performing policy. Quantitative and empirical results show that the proposed method can learn diverse skills and perform well.

**Strengths:**

- The paper tackles an interesting and important field for learning robot policies.
- Generally, the paper is well motivated. The reader can understand why the proposed method is important to the field and follow the structure to easily understand the underlying idea.

**Weaknesses:**

- For me, the usage of the term curriculum learning is unclear. The paper states that the first stage of learning diverse skills is the curriculum, however, from my understanding, the curriculum learning is to start from easier tasks and gradually increase the task complexity (see Questions).
- Although the work discusses relevant related works, it lacks a discussion of a field of works that learn diverse behaviors based on the pure maximum entropy RL framework [1,2]. How does the proposed method compare to those methods, and why is it beneficial to follow the constrained Diversity optimization framework? As also stated in the paper, the work in [2] also considers learning skills on a trajectory-level as proposed in this paper.

[1] T. Haarnoja, et al. Latent Space Policies for Hierarchical Reinforcement Learning. ICML 2018.

[2] O. Celik, et al. Acquiring Diverse Skills using Curriculum Reinforcement Learning with Mixture of Experts. ICML 2024.

**Questions:**

- How does the method generalize to varying initial positions? From my understanding, the learned skills in the first stage are related to fixed initial positions, such that for every new initial position, a set of diverse skills must first be learned. How does the proposed method react to this situation?

- From my understanding, the diversity is discovered in the trajectory level by measuring how far single features/states are away from each other. How is it guaranteed that diversity arises from this measurement? I understand that this is an effective measurement, for example, on the cylinder avoidance task, where the trajectories are represented in the XY-plane of the end-effector. However, in the case where the robot is controlled by joint trajectories, is it guaranteed to discover diversity? The robot might have redundancies that do not necessarily lead to diverse behaviors. Is it required to define a task-specific diversity measure in this case?

- Related to the prior question, how does the diversity measurement actually scale? From Eq. 3 and Eq. 7, it is clear that a set of reference trajectories needs to be saved. Although the paper states that only one reference trajectory is saved to compare the distance, this reference trajectory needs to be found by running the nearest neighbors method, which again requires calculating the distances over a set of trajectories.

- How exactly is the policy extraction done? It would be good for the reader to provide explicit objective functions for optimizing the reactive policy.

- Where exactly is the curriculum learning? Curriculum learning in RL (see e.g., in [1,2,3]) has already been proposed in the literature, also in the diverse skill discovery setting, both in RL [4] and Imitation Learning [5]. In these cases, the curriculum mostly represents tasks that change in difficulty.


 [1] C. Florensa, et al. Reverse curriculum generation for reinforcement learning. ICLR 2017.

 [2] C. Florensa. et al. Automatic goal generation for reinforcement learning agents. ICML 2018.

 [3] P. Klink, et al. Self-Paced Deep Reinforcement Learning. NeurIPS 2020.

 [4] O. Celik, et al. Acquiring Diverse Skills using Curriculum Reinforcement Learning with Mixture of Experts. ICML 2024.

 [5] D. Blessing, et al. Information maximizing curriculum: A curriculum-based approach for learning versatile skills. NeurIPS 2023.

---

### Official Review · Reviewer_Zexh · 2025-10-29

**Soundness:** 3
**Presentation:** 2
**Contribution:** 3
**Rating:** 4
**Confidence:** 4

**Summary:**

This paper addresses the challenge of discovering a diverse set of high-performing policies in complex reinforcement learning (RL) tasks, such as robotic manipulation. The authors identify a key limitation in existing constrained-diversity RL methods (e.g., Domino): they often ​​under-explore​​ in high-dimensional spaces, leading to a lack of genuine behavioral diversity as policies collapse to a few local optima. The core contribution is a novel ​​two-stage curriculum​​ that introduces a strong inductive bias for exploration. The curriculum's key idea is to decouple exploration from policy learning:

​​- Stage 1 (Trajectory-First Exploration)​​: A method called ​​Constrained Novelty Search (CNS)​​ uses an evolution strategy (CMA-ES) to optimize a population of open-loop, spline-parameterized trajectories. CNS maximizes the diversity of these trajectories (measured by a particle-based entropy estimator on state features) under a performance constraint enforced by Lagrange multipliers.

​​- Stage 2 (Policy Distillation)​​: The diverse, high-reward trajectories from CNS are used to warm-start the training of multiple skill-conditioned policies using an off-policy RL algorithm (a modified version of Domino). The paper provides specific recommendations for stabilizing this offline-to-online phase, such as symmetric sampling and a balanced update-to-data (UTD) ratio for the policy and critic networks.

Empirical evaluation on robotic tasks (cylinder avoidance, cube pushing, button pressing) demonstrates that this curriculum enables the discovery of significantly more diverse behaviors than baseline methods, including vanilla diversity optimization, action/parameter noise, DIAYN, and RND.

**Strengths:**

​​1. The two-stage "trajectory-first" approach is the primary strength. It is a conceptually clean and effective way to tackle the exploration problem in high-dimensional diversity search.

2. ​​The paper is backed by extensive experiments, including compelling qualitative visualizations (Figures 2, 3) and solid quantitative comparisons across multiple tasks and seeds (Figure 4).

3. ​​The analysis goes beyond simply proposing a new method. It offers deep insights into whyit works, through ablations on the CNS objective, the necessity of the diversity objective in stage 2, and practical training recipes (UTD ratio).

​​4. The use of spline-based trajectories is a well-justified and effective bias that makes the exploration problem tractable.

**Weaknesses:**

1. ​While the method is intuitively well-motivated and empirically validated, a more formal theoretical discussion on the guarantees of the two-stage curriculum (e.g., conditions under which trajectory-space diversity leads to policy-space diversity) would strengthen the foundation.

2.The trajectory parameterization works well for the evaluated tasks, but its scalability to problems requiring much longer horizons or more complex, hierarchical behaviors is not discussed. An analysis or discussion of the limitations regarding the dimensionality m×uof the trajectory space would be helpful.

​​3. Lack of investigation of related work. Actually, there are recent works of QD-RL that seems to address the similar problem as yours. For example: [1] and [2] both facilitate learning a diverse set of high-performing policies via reinforcement learning or inverse reinforcement learning. Acknowledgement, discussion or baseline comparison might be necessary.

[1] Wan, Z., Yu, X., Bossens, D. M., Lyu, Y., Guo, Q., Fan, F. X., ... & Tsang, I. Diversifying Robot Locomotion Behaviors with Extrinsic Behavioral Curiosity. In Forty-second International Conference on Machine Learning.

[2] Batra, S., Tjanaka, B., Fontaine, M. C., Petrenko, A., Nikolaidis, S., & Sukhatme, G. (2023). Proximal policy gradient arborescence for quality diversity reinforcement learning. arXiv preprint arXiv:2305.13795.

**Questions:**

1. ​​The choice of a B-spline for trajectory parameterization is a key inductive bias. How sensitive are the results to the choice of the number of control points m? Did you explore alternative parameterizations, and what are the theoretical or empirical guidelines for choosing mto balance expressivity and optimization efficiency for a new task?

​​2. The current CNS explores open-loop trajectories. For tasks that require long-horizon reasoning or hierarchical strategies (e.g., "grasp then place"), could the trajectory-first approach be extended? For instance, by having a higher-level CNS plan a sequence of sub-goals for lower-level policies?

​​3. The diversity objective relies on a fixed random projection of states. Did you experiment with using features learned in an unsupervised manner (e.g., from successor features or other representation learning methods) instead of the random projection? Could a learned feature space potentially capture more semantically meaningful diversity, especially in pixel-based tasks?

4. There are recent works of QD-RL that seems to address similar problem as yours. For example: [1] and [2] both facilitate learning a diverse set of high-performing policies via reinforcement learning or inverse reinforcement learning. What is the difference of settings between your work and these works? If the settings are the same, you may need to incorporate these methods as baseline for comparison. If not the same, then discussion or acknowledgement would be necessary.

[1] Wan, Z., Yu, X., Bossens, D. M., Lyu, Y., Guo, Q., Fan, F. X., ... & Tsang, I. Diversifying Robot Locomotion Behaviors with Extrinsic Behavioral Curiosity. In Forty-second International Conference on Machine Learning.

[2] Batra, S., Tjanaka, B., Fontaine, M. C., Petrenko, A., Nikolaidis, S., & Sukhatme, G. (2023). Proximal policy gradient arborescence for quality diversity reinforcement learning. arXiv preprint arXiv:2305.13795.


Glad to raise my score if these concerns are addressed.

---

### Official Review · Reviewer_HQFJ · 2025-10-31

**Soundness:** 2
**Presentation:** 3
**Contribution:** 2
**Rating:** 2
**Confidence:** 2

**Summary:**

In this paper, the authors propose a novel approach for learning diverse policies. The approach consists of two main steps. First, they generate diverse behaviors using spline-based trajectory priors, and then optimize diverse step-based policies using reinforcement learning in the second step.

**Strengths:**

This paper deals with diversity in policy learning, and that is an important topic. The approach proposes a novel combination of spline-based exploration and learning step-based policies using reinforcement learning. Overall, the method is well-supported mathematically.

**Weaknesses:**

The authors evaluated the proposed approach on only a smaller 3 tasks. Moreover, the value of diversity for two of the tasks is questionable, namely cube push and button press.

Additionally, the results presented in Figure 4, show returns of their method lower than the baselines. While diversity is useful, their method losses out on performance as it achieves lower returns. Also in ablation presented in Figure 5b, without the diversity, the method achieves better mean diversity but worse return. However, both in confidence intervals.

While the authors use step-based RL in the second phase of their approach, it would also be possible to use episode-based RL. In that case, it would be useful to compare it to existing methods, which the authors mentioned in the related works. Moreover, the authors claim that those methods have drawbacks in practical domains such as robotics. However, this claim is not further supported and if it is used to contrast with their approach, it is somewhat hard to make since the authors have not evaluated the proposed method on a real-world robotics task. This comment refers to text and works in Line 250.

The authors do not provide code.

**Questions:**

Please address the points raised in Weaknesses.

Is the Cylinder avoidance task the same as the Maze task? Based on the writing, it is, but the naming is different in Figures 2 and 4.

Have you considered diffusion models? Recent works have managed to train diffusion policies using RL, and they inherently have diverse behaviors.

---

### Official Review · Reviewer_VTZ9 · 2025-11-01

**Soundness:** 2
**Presentation:** 2
**Contribution:** 3
**Rating:** 4
**Confidence:** 3

**Summary:**

The paper proposes a two-stage method to train reinforcement learning agents that can solve a task in many different ways. In the first stage, the authors search for diverse and high-reward trajectories by optimizing smooth spline-based motion paths with an evolutionary algorithm. This produces a set of distinct ways to complete the task. In the second stage, these trajectories are used to train a single skill-conditioned policy that can reproduce and refine those diverse behaviors while still meeting performance constraints. The method, called trajectory-first constrained diversity optimization, is tested in simulated robot manipulation tasks such as obstacle avoidance, cube pushing, and button pressing. It achieves higher diversity and comparable task performance to previous diversity-driven RL approaches, demonstrating that exploring in trajectory space before policy learning can lead to richer and more varied solutions.

**Strengths:**

Reframing diversity exploration to trajectory space with B‑splines and population‑wise Lagrange multipliers is a clean, compelling bias. The  UTD recipe for constrained diversity is a practically relevant training insight.

Strong ablations substantiate design choices: Lagrangian vs scalarized novelty, with/without diversity in the RL stage. Clear comparisons to multiple good baselines

The results show substantial diversity gains on the experiments at high return, and the qualitative behaviors are convincing.

The high‑level pipeline is well conveyed; the objective and training signals are mathematically well specified.

**Weaknesses:**

All tasks are compact, deterministic simulations. The curriculum’s value for contact‑rich, stochastic manipulation and real‑world execution is not assessed.

Figures show end‑to‑end policy behaviors, but I could not find a stand‑alone characterization of CNS outputs before RL. I would like to see its performance quantitavely on all tasks and visually on avoidance task.

No code is provided, which increases reproducubility concerns.

The sensitivity of the results to alpha value should be shown in the paper.

Method section is hard to follow because of the engineering load to stabilize the training.

**Questions:**

How exactly are v* and alpha estimated per stage and over time?

Please provide performance and diversity of CNS trajectories alone (pre‑RL).

Your Stage‑1 uses spline‑parameterized open‑loop trajectories with CMA‑ES under a diversity/near‑optimality objective
Could you explicitly position this against:

Quality‑Diversity (QD) methods such as MAP‑Elites that evolve archives of high‑performing diverse solutions (often over policy or primitive parameters), why is CNS‑over‑splines preferable, and would a MAP‑Elites baseline on the same trajectory parameterization close the gap?

Movement primitives optimized by evolutionary methods, e.g., DMPs and ProMPs (classical low‑dimensional trajectory representations widely used with evolutionary search): in what ways is your spline representation and Lagrangian near‑optimality constraint genuinely different or stronger than running QD/ES directly over DMP/ProMP parameters (and then distilling)? Please consider an ablation swapping splines for DMP/ProMP under the same CNS/constraint setup.

Reward‑Conditioned Neural Movement Primitives : both methods use population‑based search. What is the key advantage of CNS over RC‑NMP’s reward‑conditioned latent‑space generation + crossover/mutation?

---

### Note · Authors · 2025-11-28

**Comment:**

We thank the reviewers for their valuable comments. We agree that the empirical evaluation of our approach should be extended, which is not possible in a satisfactory manner during the short time window of the rebuttal. We will work on improving the paper based on the reviews and resubmit an updated version thereof in the future.

**Withdrawal Confirmation:**

I have read and agree with the venue's withdrawal policy on behalf of myself and my co-authors.